# Biliary Tree Diagnostics: Advances in Endoscopic Imaging and Tissue Sampling

**DOI:** 10.3390/medicina58010135

**Published:** 2022-01-17

**Authors:** Matteo Ghisa, Angelo Bellumat, Manuela De Bona, Flavio Valiante, Marco Tollardo, Gaia Riguccio, Angelo Iacobellis, Edoardo Savarino, Andrea Buda

**Affiliations:** 1Gastroenterology Unit, Department of Oncological Gastrointestinal Surgery, S. Maria del Prato Hospital, 32032 Feltre, Italy; matteo.ghisa@aulss1.veneto.it (M.G.); angelo.bellumat@aulss1.veneto.it (A.B.); manuela.debona@aulss1.veneto.it (M.D.B.); flavio.valiante@aulss1.veneto.it (F.V.); marco.tollardo@aulss1.veneto.it (M.T.); gaia.riguccio@aulss1.veneto.it (G.R.); angelo.iacobellis@aulss1.veneto.it (A.I.); 2Gastroenterology Unit, Department of Surgery, Oncology and Gastroenterology, University of Padua, 35121 Padua, Italy; edoardo.savarino@unipd.it

**Keywords:** cholangioscopy, endoscopic retrograde cholangiopancreatography, endoscopic ultrasound, biliary tree disorders, cholangiocarcinoma, indeterminate strictures

## Abstract

The diagnostic approach to the biliary tree disorders can be challenging, especially for biliary strictures. Albeit the great diagnostic impact of endoscopic retrograde cholangiopancreatography (ERCP) which allows one to obtain fluoroscopic imaging and tissue sampling through brush cytology and/or forceps biopsy, a considerable proportion of cases remain indeterminate, leading to the risk of under/over treated patients. In the last two decades, several endoscopic techniques have been introduced in clinical practice, shrinking cases of uncertainties and improving diagnostic accuracy. The aim of this review is to discuss recent advances and emerging technologies applied to the management of biliary tree disorders through peroral endoscopy procedures.

## 1. Introduction

Despite the first visualization of the biliary tract mucosa having happened a long time ago and technological advances have dramatically increased over the last five decades (Figure 1), biliary tree disorders remain, in some cases, a diagnostic challenge due to the low accuracy of conventional approaches (Figure 1).

Clinical presentation is extremely variable, from asymptomatic accidental finding during imaging performed for other reasons, to abnormal liver tests, or symptoms/signs as jaundice, pruritus, abdominal pain, weight loss, and asthenia [1]. Clinical evaluation and laboratory findings are usually followed by biliary imaging, as ultrasound (US) and, more frequently, magnetic resonance (MR) and/or computed tomography (CT) [1].

A great variety of biliary disorders that are either benign or malignant can be diagnosed, as shown in Table 1.

In the *mare magnum* of possible findings in the biliary tract strictures appears to be the most challenging.

Biliary strictures are considered indeterminate when the traditional management with radiologic evaluation and endoscopic retrograde cholangiopancreatography (ERCP) plus brush cytology and/or forceps biopsy does not lead to a diagnosis [1].

There is a wide range of findings, from benign to malignant conditions (Table 2), with inflammatory strictures linked to stones not infrequent among the formers. Other conditions such as primary sclerosing cholangitis (PSC), recurrent infectious cholangitis, and IgG4-related cholangitis, previous surgery, as cholecystectomy or liver transplantation, can also cause benign biliary strictures.

Moving to malignant strictures, pancreatic cancer and cholangiocarcinoma should be always ruled out with certainty. Pancreatic adenocarcinoma usually determines distal common bile duct stenosis, while cholangiocarcinoma could involve all the biliary ducts, both intra- and extrahepatic. Other malignancies can cause biliary strictures, due to anatomical contiguity, as cancer of the ampulla of Vater, gallbladder, liver parenchyma, and metastatic lesions.

Up to 20% of biliary strictures remain indeterminate after the standard approach and in the past these patients underwent surgery to achieve a definitive diagnosis. Recently, evaluation of these strictures remains challenging and often requires a multidisciplinary approach and multiple procedures. However, new weapons have become available. Peroral cholangioscopy allows direct visualization of these lesions and targeted tissue acquisition, using miniaturized biopsy forceps. Moreover, advanced technologies enabling visual analysis of biliary mucosa and the evaluation of histological samples have been recently developed to improve the diagnostic yield for indeterminate biliary strictures.

Learn how to navigate in this complex and intricate field of endoscopy is mandatory for the proper management of these patients and to avoid under- or overtreatment errors that negatively impact patients’ care and prognosis.

## 2. Techniques and Methodologies

### 2.1. Endoscopic Retrograde Cholangiopancreatography

Since its introduction in 1968, ERCP has become essential for the management of biliary tree disorders. Although its use has changed from a diagnostic to a therapeutic modality, ERCP still has an important role in diagnosis of biliary strictures. Features suggesting a malignant phenotype include length >1 cm, asymmetry, irregular borders or abrupt change in caliber, intraductal nodules or polypoid masses, and simultaneous dilation of both the common bile duct and the pancreatic duct (double duct sign) [2,3]. Instead, short, regular, and symmetric strictures support the diagnosis of a benign disease. These cholangiographic findings have a diagnostic sensitivity and specificity of 74% and 70%, respectively [4].

Efforts to improve the accuracy of cholangiographic findings resulted in methods that allow cyto- and/or histological confirmation. Bile juice aspiration, brush cytology, and forceps biopsies are the main ways to acquire tissue during ERCP. Although bile aspiration cytology is easy to perform, it is burdened by low sensitivity (6–24%) [5,6]. Better diagnostic performances can be reached with brushing cytology, with a sensitivity and specificity for malignancy ranging between 21% and 70% and 97% and 100%, respectively [7]. In a review considering 16 studies, the overall biliary brush cytology sensitivity was 42% with a negative predictive value (NPV) of 58% [8]. Low values of sensitivity were firstly related with sampling errors or inadequate specimens, but also difficulties in cytopathologic distinction between malignant and non-malignant cells were considered [9,10]. While the use of longer brushes or preventive stricture dilation do not clearly increase the diagnostic sensitivity, repeated brushing seems able to improve accuracy [11]. Fluoroscopic guided forceps biopsies allow to obtain histological diagnosis though resulting more technically difficult and time consuming. Studies suggest that at least 2–3 specimens should be acquired [12,13] with a reported sensitivity ranging between 43% and 81% and a specificity of 90–100% [7]. A meta-analysis found the pooled sensitivity and specificity of brush cytology of 45% and 99%, respectively, compared to 48.1% and 99.2% obtained by intraductal biopsy in the evaluation of biliary strictures; the combination of both modalities improved the sensitivity to only 59% [14]. PSC represents a critical field, given the increased risk of cholangiocarcinoma (CCA) compared to the general population and the high occurrence of stenosis in these patients. Again, low sensitivity, ranging from 18% to 40%, has been shown in these patients [15,16,17]. Difficulties in gaining access and obtaining samples from the target area, possible submucosal growth of CCA, and desmoplastic reaction are some of the reasons for this poor diagnostic ability [18].

Concerning safety data, it should always be considered that ERCP is an invasive procedure and this concept has changed over years, as the main goal of ERCP has changed from being a diagnostic technique to a therapeutic one. The rates of pancreatitis, cholangitis, and perforation have been reported to be 3.5–9.7%, 0.5–3.0%, and 0.08–0.6%, respectively, with a mortality rate of post-ERCP pancreatitis of 0.1–0.7% [19,20,21,22].

### 2.2. Cholangioscopy

Cholangioscopy can be performed via the percutaneous transhepatic or peroral routes and it allows to access and visualize the biliary tree. Diagnostic cholangioscopy is mainly required for the evaluation of biliary strictures but can also be useful to discern etiology of unexplained filling defects at fluoroscopy and abnormalities detected during CT, MR, and endoscopic ultrasound (EUS) [23,24]. Moreover, its application can help defining longitudinal extension of CCA more accurately than the fluoroscopic measurement during ERCP [25].

The percutaneous transhepatic approach requires the creation of a track placed by interventional radiology. It is invasive and associated with long hospitalization and high risk of complications. Thus, it has been almost abandoned and the oral route is now preferred [26,27].

Indeed, with the peroral cholangioscopy the biliary tree is achieved through the patient’s mouth. It can be performed via direct advancement of an endoscope into the bile ducts or indirectly, with the so called “mother-baby system”, where a cholangioscope is entered through the operative channel of a duodenoscope.

Direct peroral cholangioscopy (DPC) is performed by using an ultra-slim forward-viewing endoscope, advanced directly into the papilla as during a regular endoscopy. Alternatively, the ultra-slim endoscope can be loaded over a guidewire previously located into the bile ducts during ERCP. The application of an overtube allows for easier reaching of the papilla and preventing looping of the endoscope in the stomach. Other approaches require the use of a balloon catheter loaded over a guidewire, inflated, and anchored into the bile duct [28,29,30]. ERCP with sphincterotomy must be previously performed in order to achieve successful duct access through the papilla. The reusable guiding probe of Katz which is similar to a papillotome is another device that can be used. After its insertion into the bile duct and the removal of the duodenoscope, an ultra-slim endoscope can be advanced over the probe and gain the biliary tree [31]. The most difficult aspect of the direct cholangioscopy is the opposite axis of the common bile duct compared to the upper GI tract. To overcome this problem, a double bending cholangioscope has been developed [32]. Thanks to the presence of two bending sections, proximal and distal, this cholangioscope makes the direct evaluation of the biliary tree easier. In a series of 41 patients undergoing direct cholangioscopy, a guide wire or balloon assisted technique resulted in a technical success in 88.2% of cases [33]. In a more recent study, 74 patients have been evaluated with a newer prototype of this cholangioscope; the papilla was entered in 97% cases, most of them with a freehand technique, and the targeted bile duct was evaluated in 84% of patients [34]. Despite clear advantages (i.e., superior image quality and larger operative channel), DPC is burdened by several limitations that avoid widespread use. First, it is technically challenging with difficulties in the advancement of the endoscope due to gastric loops and low stability of the instrument into the bile ducts. In contrast, the balloon assisted technique does not allow the use of accessories as the working channel is occupied. Second, these procedures remain time consuming, limited to experienced endoscopists and have a high rate of failure. Due to these limitations, indirect peroral cholangioscopy (IPC) has gained worldwide popularity over the direct approach. Traditionally, IPC was performed with the dual-operator system, in which the cholangioscope (baby) is advanced into the bile ducts through the working channel of the duodenoscope (mother). The procedure requires two endoscopists, one operating each endoscope. Due to the high instrument and staff related costs, the fragile structure and the time consuming process, this technique is no longer available [35]. IPC is now performed with a single endoscopist operating both the cholangioscope and the duodenoscope, with the former introduced into the working channel of the latter and then advanced into the biliary tree over a guide wire. As for DPC, sphincterotomy is usually performed before cholangioscope introduction. The first-generation of these cholangioscopes was launched in 2007 with a single-use fibro-optic-based device (SpyGlass, Boston Scientific, Marlborough, MA, USA) [36]. In 2015, a newer generation of cholangioscope (SpyGlass DS, Boston Scientific) has been introduced and now the third series (SpyGlass DS II, Boston Scientific) is on the market. The SpyScope consists of a sterile, completely disposable delivery catheter, connected to the SpyGlass digital controller. This system has an improved, high-quality, digital imaging system and provides a wider endoscopic field of view. The SpyScope catheter consist of a 10 Fr (3.3 mm) outer diameter, 230-cm length catheter. A dedicated 1.2-mm accessory channel allows the passage of dedicated accessories to perform biopsy, endomicroscopy, laser or electrohydraulic lithotripsy, and snare or basket retrieval [37,38]. In a recent study, digital cholangioscope had better image quality, visualization, and maneuverability than previous semidisposable, fiberoptic cholangioscopes [39].

Cholangioscopes allow one to visualize and characterize the mucosa and recognize patterns that enable one to discern benign from malignant lesions. Features related to malignant strictures are irregular, dilated and tortuous vessels, nodularity, neovascularization, easy oozing, and papillary or granular irregular surfaces (Figure 2). In contrast, benign findings include smooth mucosa, fine network of thin vessels, borders without neovascularization, and homogeneous papillo-granular surface with no masses (suggesting hyperplasia), bumpy surface with or without pseudodiverticula (suggesting inflammation), and white surfaces with a convergence of folds (suggesting scarring) [40,41,42]. A recent review provided data on the diagnostic yield of cholangioscopy for indeterminate biliary strictures based on visual findings. The sensitivity for malignancy is in the range of 83–100%, the specificity 67–96%, and the accuracy 85–96% [7]. These findings have been confirmed by a meta-analysis showing a sensitivity and specificity of 84.5% and 82.6%, respectively [43]. The use of digital cholangioscopes further improved the diagnostic yield compared to the first generation of fibro-cholangioscopes. A multicenter study evaluating 44 patients, reported a sensitivity and specificity of 90% and 95.8%, respectively, for visual diagnosis of malignant biliary strictures [37]. Next, a prospective long-term follow up study with SpyGlassDS, showed a sensitivity and specificity of visual diagnosis of 95.5% and 94.5%, respectively [44]. However, visual criteria for malignancy are not fully established and uncertainties remain along with a significant interobserver variability that limit the accuracy of these stigmata. Thus, a visual diagnosis remains unfeasible at least in some cases and histological evaluation is indeed frequently required to have a confirmation. A higher accuracy for IPC guided biopsies with mini forceps compared to fluoroscopic guided standard forceps biopsy, has been reported in a prospective study [45]. Accordingly, in a systematic review and meta-analysis the diagnostic yield of IPC-guided biopsies in determining malignant biliary strictures showed a sensitivity and specificity of 60.1% and 98.0%, respectively [43]. In a more recent review, an IPC-guided biopsy for indeterminate biliary strictures showed good diagnostic yield with a sensitivity ranging between 64% and 86%, a specificity of 89–100%, and an accuracy of 70–90% [7]. Despite great advances achieved by the introduction of Spyglass systems, performing IPC to evaluate distal bile duct strictures is challenging. Limited bending property of the cholangioscope tip makes it difficult to target the biopsy site of interest in distal bile duct lesions with lower adequate tissue acquisition compared to proximal strictures [46].

Bearing in mind that IPC is at least as invasive as ERCP, adverse events (AEs) should always be considered. A systematic review and meta-analysis of clinical studies including different type of oral cholangioscopy showed an overall and serious AEs rates of 7% and 1%, respectively, with cholangitis being the most common AE (4%) [47]. In a study including only the single operator cholangiopacreatoscopy, AEs range between 2% and 30%, with cholangitis, pancreatitis, hemobilia, and bile leak being the most frequently reported items. Perforation is a rare but serious AE that can occur. Considering DPC, air embolism is another rare AE, reported in 0–2.3% of procedures. Water or CO_2_ use instead of air is recommended to minimize this fatal AE [48]. Sethi et al. compared the AEs rate of patients who underwent (*n* = 402), or not (*n* = 3475), cholangioscopy during ERCP. A higher rate has been reported in the first group (7% vs. 2.9%, respectively; OR 2.5 CI 1.56–3.89) [49]. The subgroup analysis revealed a higher rate of cholangitis (1.0% vs. 0.2%, respectively), with similar rates of pancreatitis and perforation [49].

As cholangitis is clearly the most frequently reported AE, a prophylactic peri-procedural antibiotic therapy can significantly reduce this risk and indeed it is recommended in all patients [44].

### 2.3. Narrow Band Imaging

Narrow Band Imaging (NBI) allows one to obtain enhanced images. The white light is filtered; thus, eliminating all wavelengths except for wavelengths absorbed by hemoglobin, for maximum contrast. Indeed, it provides better visualization of microvessels and mucosal pattern. Its usefulness during standard endoscopy is beyond any doubt, but several studies have also reported the value of its application during cholangioscopy [50,51,52,53]. NBI increased the biopsy rate in patients with PSC although NBI directed biopsies did not improve the dysplasia detection rate [54]. In another series of 38 patients with indeterminate strictures, NBI increased the ability to detect malignancies in 89.4% by enhancing the mucosa pattern [55,56]. In another study, NBI during cholangioscopy results were superior to MR in accurately determining hepatobiliary tumor margins, as recently confirmed during surgical resection [57]. Although NBI significantly improves the diagnostic ability of cholangioscopy, it remains an optical technique that is only able to assess superficial findings. Submucosal spread or extraductal invasion of the tumor cannot be evaluated by cholangioscopy with NBI.

### 2.4. Dye Chromoendoscopy

Chromoendoscopy represents the gold standard for the identification of dysplastic areas during standard endoscopy in many precancerous conditions such as inflammatory bowel diseases and Barrett esophagus [57,58]. Methylene blue application during cholangioscopy can enhance subtle changes of mucosal pattern of the bile ducts. In a study on 45 biopsy specimens obtained during cholangioscopy, the cancerous epithelia stained significantly less often (0%) than either the normal (90%, *p* < 0.001) or the metaplastic (69%, *p* = 0.001) epithelia [59]. In a group of patients that underwent chromocholangioscopy for biliary strictures, homogenous light blue staining was associated to normal epithelium whereas heterogeneous intense dark staining was seen in inflammatory and dysplastic lesions. Weak staining was also identified in strictures related to PSC and post-liver transplant [60]. However, methylene blue can also adhere to mucin and exudates, reducing accuracy and requiring expertise in image interpretation [61].

### 2.5. Endoscopic Ultrasound

EUS allows an excellent comprehensive evaluation of the hepato-pancreato-biliary system. EUS with or without fine needle aspiration (FNA) can differentiate malignant from benign biliary strictures with a greater diagnostic performance in distal common hepatic duct strictures compared to proximal and intrahepatic ones. Findings such as pancreatic head mass with secondary extrinsic compression of the bile duct, thickening >3 mm or irregular outer edge of the bile duct wall have been linked to malignant biliary strictures. In a meta-analysis published in 2007, nine studies including 555 patients were evaluated: EUS without FNA diagnosed malignancies of the biliary tree with a sensitivity and specificity of 78% and 84%, respectively [62]. When assessed with other imaging techniques, cholangiocarcinoma detection was higher by EUS compared to CT and MR (94%, 30%, and 42%, respectively). However, new advancements in imaging accuracy and operator dependency of EUS have to be considered and therefore these data should be considered with caution [63]. In the subset of patients with asymptomatic common bile duct (CBD) dilation, EUS is increasingly used in the diagnostic work up. Chooda and coworkers recently performed a systematic review and meta-analysis evaluating EUS yield in these patients; eight studies including 224 patients, were considered and the cumulative yield of EUS for any pathology was 11.2% (95% CI, 3.6–21.6%). The EUS yield for benign etiologies was 9.2% (95% CI, 1.1–21.9%) among which choledocholithiasis comprised 3.4% (95% CI, 0–11.2%) and 0.5% (95% CI, 0–3.4%) for malignant etiologies. This low but not insignificant rate raises cost-effectiveness issues that should be addressed with patients in clinical decision-making [64].

FNA during EUS is a minimally-invasive option to obtain tissue sampling for cytological diagnosis and although its main application is on pancreatic masses, it is successfully applied to biliary lesions [63,65,66,67,68,69]. De Moura et al. performed a meta-analysis comparing ERCP (with brush cytology or forceps biopsy) and EUS–FNA diagnostic yield for malignant biliary strictures. A total of 294 patients were included, the mean sensitivities of ERCP and EUS–FNA for tissue diagnosis of malignancy were 49% and 75%, respectively; the specificities were 96.33% and 100%, respectively [70]. More recently, data on the diagnostic yield of EUS–FNA for biliary tract malignancy has been summarized in a review, with a sensitivity ranging between 43–94%, a specificity of 100%, and an accuracy of 70–94% [7]. Interestingly, the site of the biliary tree involved seems to play an important role. In one study, the sensitivity for extrahepatic CCA was significantly higher than that for proximal CCA (81% vs. 59%) [63].

EUS fine-needle biopsy (FNB) yielded similarly or even better than FNA for tissue evaluation of several different solid lesions, such as pancreatic and submucosal ones, overcoming sampling material inadequacy and decreasing the number of needle passages needed to obtain adequate material [71]. A recent meta-analysis compared FNA with FNB, indicating that FNB provided a higher diagnostic accuracy, in both pancreatic and non-pancreatic lesions [72]. Data on the yield of FNB for biliary tree malignancies are lacking, but an improvement in sensitivity could be conceivable.

Concern for potential seeding of malignant cells along the needle track remains a grey area. In a series of 191 patients with hilar CCA receiving neoadjuvant chemoradiation followed by liver transplantation, transperitoneal biopsy of hilar cholangiocarcinoma was associated with a higher rate of peritoneal metastases (*p* < 0.01). Subsequently, authors suggested to avoid FNA when a liver transplant approach for CCA, is available [73]. It is not clear whether this concern should be transferred from percutaneous to the trans-duodenal FNA approach. A retrospective study showed that preoperative EUS–FNA in patients with CCA did not affect overall or progression-free survival [74]. Indeed, while additional studies will help understanding this area of uncertainty, biliary specimens to rule out CCA should be acquired intraductally rather than transmurally when liver transplantation is taken into consideration.

### 2.6. Intraductal Ultrasound

Intraductal ultrasound (IDUS) involves a thin (2.0–3.1 mm), high-frequency (12–30 MHz), wire-guided ultrasound miniprobe that can be advanced into the bile duct through the working channel of the duodenoscope. It allows a better resolution compared to EUS due to direct contact with biliary mucosa and the higher frequency. Although it is rarely used in routine clinical practice, IDUS can be helpful in the evaluation of biliary strictures. The finding of disruption of the three-layer architecture of a normal bile duct, eccentric wall thickening, hypoechoic mass with irregular margins, papillary surface, and malignant-appearing periductal lymph nodes, are among the features linked to malignancy [75]. In a retrospective study by Meister, 397 patients with indeterminate bile duct strictures were included; 264 were finally referred to surgery due to high suspicion of malignancy and 20 benign cases were misclassified by IDUS as malignant, whereas 14 malignant strictures were initially reported as benign. Sensitivity, specificity, and accuracy rates of IDUS were 93.2%, 89.5%, and 91.4%, respectively. In the subgroup analysis of malignancy prediction, IDUS showed the best performance in cholangiocellular carcinoma with a sensitivity of 97.6% [76]. Data from other groups showed that the additional use of IDUS to ERCP plus tissue sampling is able to improve sensitivity from 41–68% to 90–93% [77,78,79,80]. Menzel et al. compared the accuracy of IDUS and EUS in diagnosing biliary obstruction and in predicting surgical resectability; IDUS exceeded EUS in terms of accuracy (89.1% vs. 75.6%; *p* < 0.002), sensitivity (91.1% vs. 75.7%; *p* < 0.002), specificity (80% vs. 75% *p* = NS), and T-staging (77.7% vs. 54.1%; *p* < 0.001). However, IDUS understaged lymph node staging particularly in pancreatic tumors (IDUS 13.3% vs. EUS 69.2% correct *p* < 0.002) with an overall accuracy of 33.3% [81]. Consistently a single center study on cancer staging reported IDUS accuracy of 69% for of N0 and N1, confirming that the limited depth of ultrasonic penetration represents a major drawback of IDUS tumor staging [76,80]. In a retrospective study on 193 patients with biliary obstruction, IDUS correctly identified 94 of 97 (97%) malignancies and 76 of 96 (79%) benign conditions with sensitivity, specificity, and accuracy rates of 96%, 79%, and 88%, respectively [82]. The accuracy rate of IDUS was higher for proximal compared to distal bile duct obstruction (98 vs. 83%, *p* = 0.006). Biliary wall thickness >7 mm without extrinsic compression had a positive predictive value (PPV) of 100% for including malignancy, whereas length ≥20 mm demonstrated a PPV of 93% [82].

IDUS application has been expanded into other clinical conditions. IDUS findings of circular-asymmetric wall thickness, irregular inner margin, diverticulum-like outpouching, and disappearance of the three layers were able to differentiate IgG4 related sclerosing cholangitis to PSC [83]. In portal biliopathy, IDUS can discriminate biliary narrowing secondary to pericholedochal collaterals from strictures [84,85]. Moreover, it has been shown that IDUS can be feasible and effective in performing ERCP without contrast cholangiography [86].

### 2.7. Confocal Laser Endomicroscopy

Confocal laser endomicroscopy (CLE) is an endoscopic imaging technique able to gain histological assessment in real-time, being often termed “virtual biopsy”.

The CLE imaging probe (CholangioFlex, Cellvizio; Mauna Kea Technologies, Paris, France) has a lateral resolution of 3.5 μm and a diameter of 0.94 mm. It can be used under fluoroscopy guidance or directly during cholangioscopy. CLE miniprobes can be accurately applied to the target tissue providing highly accurate and sensitive characterization of biliary stricture [87,88]. CLE requires administration of intravenous or topical contrast (typically fluorescein) to highlight tissue features and better differentiate normal architecture or inflammatory changes from neoplastic tissue.

The Miami classification harvested criteria for the identification of malignancy, including thick white (>20 mm) or dark (>40 mm) bands, dark clumps, or epithelial structures. Finding of any of these criteria yielded a high sensitivity of 97% but low specificity of 33%, due to the high rate of false-positive results. Moreover, this expert consensus based classification has shown a low interobserver agreement [89]. Subsequently, the Paris classification was proposed to identify inflammatory conditions based on four findings: dark granular patterns with scales, increased interglandular space, vascular congestion, and thickened reticular structure [90]. In patients with indeterminate biliary strictures, CLE has shown greater sensitivity compared to ERCP with tissue sampling for diagnosing biliary cancer (98% vs. 45%) [88]. Adding CLE to ERCP plus tissue sampling has been shown to increase diagnostic accuracy on biliary strictures. In a prospective multicenter study tissue sampling alone was 56% sensitive, 100% specific, and 72% (95% CI, 63–80%) accurate. Adding CLE to ERCP, sensitivity, specificity, and accuracy increased to 89%, 71%, and 82% (95% CI, 74–89%), respectively. With the subsequent tissue sampling evaluation, strictures resulted distinguishable with 88% (95% CI, 81–94%) accuracy [91]. A recent meta-analysis on the application of CLE in biliary strictures showed pooled sensitivity and specificity of 90% and 75%, respectively [92].

Although not routinely and extensively used, CLE represents an effective tool for the diagnosis of malignant biliary strictures [93], as recently reported in the American Society for Gastrointestinal Endoscopy (ASGE) guidelines for the management of biliary neoplasia [94].

### 2.8. Autofluorescence Imaging

The principle of autofluorescence imaging (AFI) is founded on the interaction between light and tissue fluorophores. The endogenous fluorophores (e.g., porphyrins), when exposed to specific wavelength light, become excited, emitting fluorescent light (i.e., autofluorescence). Two hypersensitive cameras, with red and green spectra, then capture this light. Normal tissue usually emits green light whereas neoplastic tissue dark green or black light [95]. AFI evaluation of 65 biliary lesions was able to improve sensitivity from 88% to 100% compared to conventional cholangioscopy, whereas the specificity was reduced from 87.5% to 52.5%, due to false-positive results related to the presence of granular non-neoplastic mucosa and the dark appearance of bile [96]. This methodology is not widespread and it is rarely used.

### 2.9. Optical Coherence Tomography

Optical coherence tomography (OCT) is a real-time optical imaging technique, similar to ultrasonography, except for the use of infrared light instead of high-frequency sound waves. Light is reflected or scattered from tissues to generate a cross-sectional tomographic image of the target area [97,98]. OCT provides greater spatial resolution than IDUS and allows to evaluate larger surfaces areas compared to CLE, that also requires contrast medium injection. The second-generation of OCT system that is now on the market offers great resolution with high image quality and spatial resolution [99]. Thus, OCT resolution of layers’ architecture is comparable to histologic sections [100,101]. Different light-backscattering patterns have been reported from normal and malignant structures [102,103,104,105]. Thanks to its ability to visualize microscopic structures, OCT increases diagnostic sensitivity for malignant biliary strictures when added to biliary brushing cytology. Two OCT criteria for malignancy (unrecognizable layer architecture and presence of large non-reflective areas) were evaluated in a cohort of 37 patients with biliary strictures, and 19 patients had a diagnosis of malignancy. The two OCT above mentioned criteria showed a sensitivity, specificity, PPV, NPV, and accuracy of 53%, 100%, 100%, 64%, and 74%, respectively. A significant improvement of sensitivity from 67% to 84%, has been obtained adding OCT to standard tissue sampling [106]. The diagnostic capacity of OCT was prospectively evaluated in 12 patients with strictures of the main pancreatic duct and half of the patients (*n* = 6) finally were diagnosed with malignancy. The accuracy of OCT was higher than brush cytology (100% vs. 66.7%) with the same specificity (100%). To note, in one patient neither OCT nor brush cytology was done due to the severity of the stricture [107].

Although all of these limited existing data are encouraging, further studies are needed to understand the potential role and feasibility of OCT in clinical practice.

## 3. Conclusions

To date, ERCP with tissue sampling with brushing and/or forceps biopsy is widely practiced to obtain diagnosis and clarify biliary tree findings doubtful for malignancies. Although this approach is still considered the gold-standard, too often is non-diagnostic, delaying the decision-making process and leading to inaccurate management of these patients. Thus, it is mandatory to spread, standardize, and routinize the use of other more innovative techniques with the aim to improve diagnostic yield. New digital IPC and EUS with associated techniques for tissue sampling, should always be part of the available armamentarium in tertiary centers dealing with biliary tree disorders. In contrast, less widespread techniques such as CLE and OCT are still burdened by low levels of evidence to be considered as part of a standard diagnostic algorithm. At the same time, although it is beyond the scope of this review, we must bear in mind the importance of advancements in molecular pathology techniques, which can dramatically increase our diagnostic accuracy. With all of these new technologies available, we should be confident that the near future will see a great step forward in the endoscopic management to biliary tree disorders.

## Figures and Tables

**Figure 1 medicina-58-00135-f001:**
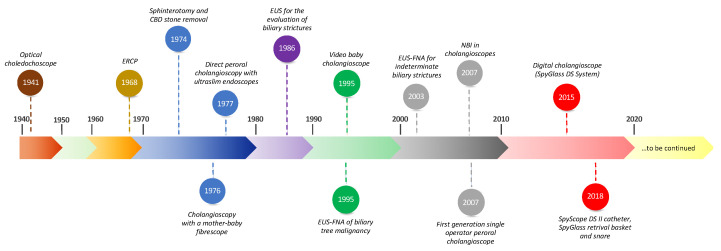
Timeline with the landmark advancements in the assessment of the biliary tree disorders.

**Figure 2 medicina-58-00135-f002:**
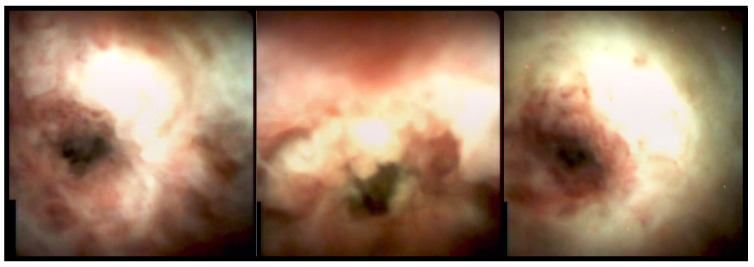
Cholangioscopy images of a stenosis just below the hepatic hilum, obtained with the SpyGlass DS system. Cholangioscopy-targeted biopsies confirmed the presence of adenocarcinoma (type I cholangiocarcinoma according to Bismuth–Corlette classification).

**Table 1 medicina-58-00135-t001:** List of biliary tree disorders commonly found in clinical practice.

Biliary Tree Disorders	
Biliary anatomical abnormalities	Choledochal Cyst, Biliary atresia, Caroli disease
Cholangitis	Infective, Primary sclerosing cholangitis (PSC), Primary biliary cholangitis (PBC), IgG4-related cholangitis, Secondary sclerosing cholangitis
Benign Tumors of the Bile Ducts	Biliary hamartoma, Intraductal papillary neoplasm of the bile duct, Biliary Mucinous Cystic Neoplasms
Malignant Tumors of the Bile Ducts	Cholangiocarcinoma
Choledocholithiasis	
Post inflammatory biliary strictures	
Disorders of gallbladder	Cholecystitis, Cholelithiasis, Mirizzi’s syndrome
Disorders of the ampulla of Vater	Ampullary tumours (adenoma and adenocarcinoma)
Iatrogenic disorders	Accidental surgical ligation
Others	AIDS cholangiopathy, parasites

**Table 2 medicina-58-00135-t002:** Potential diagnoses of indeterminate biliary stricture.

**Benign**	**Iatrogenic**	Post-surgery (e.g., cholecystectomy-related, anastomotic, ischemic)		
		Chemotherapy, radiotherapy, traumatic injury		
	**Inflammatory or Autoimmune**	IgG4-associated cholangitis PSC	Eosinophilic cholangitis Sarcoidosis	
	**Intraluminal Obstruction or Extraluminal Compression**	Choledocholitiasis Chronic pancreatitis	Mirizzi syndrome Groove pancreatitis	Inflammatory pseudotumors
	**Vascular**	Vasculitis Ischemic cholangiopathy	Portal hypertensive biliopathy	
	**Infectious**	Bacterial (recurrent pyogenic cholangitis) Parasite	Human Immunodeficiency Virus (HIV) cholangiopathy	
**Malignant**	**Pancreatic Adenocarcinoma Cholangiocarcinoma**	Ampullary adenocarcinoma Gallbladder cancer	Hepatocellular carcinoma (HCC) Lymphoma	Metastatic adenocarcinoma Compressive lymphadenopathy

## Data Availability

Not applicable.

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
