# Peer review of "Biliary Tree Diagnostics: Advances in Endoscopic Imaging and Tissue Sampling"

_medicina, 2022, doi:10.3390/medicina58010135_

Round 1
Reviewer 1 Report
Congratulations to this nice review regarding diagnostic modalities in biliary tree disorders, which is often challenging in the daily clinical setting. I have only minor concerns/ additional ideas:
- stenoses of the distal DHC are not easy to examine with spyscope DS due to instabile scope positioning in this location
- the significance of EUS is limited in proximal DHC stenosis and intrahepatic bile duct disorders, while this modality has good diagnostic accuracy in the distal DHC/pancreatic head
- I would recommend addition of intraprocedural findings (imaging) showing pathologies (e.g., malignant stenosis in cholangioscopy)
Author Response
Reviewer 1.
We are delighted that the reviewer found our review on diagnostic modalities in biliary tree disorders nice and we thanks for the additional ideas which we have addressed as follow:
- stenoses of the distal DHC are not easy to examine with spyscope DS due to instabile scope positioning in this location.
We thank the reviewer and we agree with his/her comment. We have now amended the text accordingly (highlighted in yellow in the revised version of the manuscript) and included relevant reference. - the significance of EUS is limited in proximal DHC stenosis and intrahepatic bile duct disorders, while this modality has good diagnostic accuracy in the distal DHC/pancreatic head
We thank the reviewer for the comment. We agree with the reviewer that EUS diagnostic performance is greater in distal common hepatic duct strictures compared to proximal and intrahepatic ones. We have now included this important point in the EUS section, highlighted in yellow in the revised version of the manuscript. - I would recommend addition of intraprocedural findings (imaging) showing pathologies (e.g., malignant stenosis in cholangioscopy)
We thank the reviewer for the opportunity of improving our manuscript adding cholangioscopy images showing the mucosal pattern of a malignant (histologically confirmed) stricture. We have included 3 images in Figure 2.
Reviewer 2 Report
My assessment is that the overall quality of this article is excellent and should be accepted for publishing with high priority.
The authors discuss in this review the recent advances and emerging technologies applied to the management of biliary tree disorders through peroral endoscopy procedures.
The title correctly reflects the main subject of the manuscript and the article adequately describes the background, present status and significance of the study.
Although ERCP with tissue sampling is still considered the gold-standard and is widely practiced to obtain diagnosis and clarify biliary tree findings doubtful for malignancies, it is too often non-diagnostic, delaying the decision-making process and leading to inaccurate management of these patients. Thus, the authors conclude that it is mandatory to spread, standardize and routinize the use of other more innovative techniques (like indirect peroral cholangioscopy or endoscopic ultrasound) with the aim to improve diagnostic yield.
In my opinion, this is a relevant and interesting review and having all these new technologies available we should be confident that the near future will see a great step forward in the endoscopic management of biliary tree disorders.
The manuscript discuses these recent advances and emerging technologies adequately and appropriately, highlighting the key points concisely, clearly and logically and the findings are stated in a clear and definite manner. The discussions are accurate and clear and the conclusions are consistent with the evidence and arguments presented.
The article is well, concisely and coherently organized and presented. The style, language and grammar is accurate and appropriate.
My assessment is that the overall quality of this article is excellent and should be accepted for publishing with high priority.
Author Response
Reviewer 2.
We are delighted that the reviewer found the overall quality of our review article excellent and suitable for being published with high priority. Indeed, we would like to thank the reviewer for his supportive comments and highlighting the strengths of our review article on this rapidly evolving topic.